# The Soil Bacterial Community Structure in a *Lactarius hatsudake* Tanaka Plantation during Harvest

**DOI:** 10.3390/microorganisms12071376

**Published:** 2024-07-05

**Authors:** Airong Shen, Yun Tan, Baoming Shen, Lina Liu, Jilie Li, Zhuming Tan, Liangbin Zeng

**Affiliations:** 1College of Life Sciences and Technology, Central South University of Forestry and Technology, Changsha 410004, China; shyy123@hnlky.cn (A.S.); tany@hnlky.cn (Y.T.); lijilie12@163.com (J.L.); 2Hunan Academy of Forestry, Changsha 410004, China; shenbm@hnlky.cn (B.S.); omphalina@outlook.com (L.L.); 3Institute of Bast Fiber Crops, Chinese Academy of Agricultural Sciences, Changsha 410205, China

**Keywords:** *Lactarius hatsudake* Tanaka, mycorrhizal edible fungi, bacterial community, soil microbial diversity

## Abstract

*Lactarius hatsudake* Tanaka is a mycorrhizal edible mushroom with an appealing taste and rich nutrition. It is also a significant food and has medicinal value. In this study, the plantation of *L. hatsudake* during the harvest period was taken as the research object, and this article explores which bacteria in the soil contribute to the production and growth of *L. hatsudake*. The soil of the control (CK) and the soil of the mushroom-producing area [including the soil of the base of the mushroom (JT) and the mycorrhizal root soil (JG)] was collected in the plantation. The three sites’ bacterial community structure and soil diversity were analyzed using high-throughput sequencing technology, and a molecular ecological network was built. Soil bacteria in the *L. hatsudake* plantation had 28 tribes, 74 classes, 161 orders, 264 families, 498 genera, and 546 species. The dominant phyla were Proteobacteria and Acidobacteria, and the dominant genera were *Burkholderia_Caballeronia_Paraburkholderia*, *Acidothermus*, *Bradyrhizobium*, *Candidatus_Xiphinematobacter*, and *Granulicella*. The α-diversity of soil bacteria in JT was significantly lower than that in JG and CK, and the β-diversity in JT samples was significantly different from that in JG and CK samples. The size and complexity of the constructed network were smaller in JT samples than in JG and CK samples, and the stability was higher in JT samples than in JG and CK samples. The positive correlation between species in JT samples was dominant. The potential mycorrhizal helper bacteria (MHB) species of *L. hatsudake* was determined using correlation and differential group analysis. The results support future research on mycorrhizal synthesis, plantation management, and the function of microorganisms in the soil rhizosphere of *L. hatsudake*.

## 1. Introduction

*Lactarius hatsudake* Tanaka [1,2,3] belongs to the phylum Eumycota, subphylum Basidiomycotina, class Hymenomycetes, order Agaricales, family Russulaceae, and genus *Lactarius*. *Lactarius hatsudake* is an edible mycorrhizal mushroom that is symbiotic with Pinaceae plants. *Lactarius hatsudake* is appealing and nutritious and has high edible and medicinal value. Its fruiting body is rich in crude fiber, polysaccharides, protein, vitamins, and amino acids [1,2,4]. This mushroom is effective against tumors and viruses, scavenging free radicals, improving immunity, being beneficial for the stomach and intestines, helping treat diabetes, and other medical effectiveness [5,6]. It is an ideal natural, multifunctional food that is used by consumers all over the world [7]. *Lactarius hatsudake* has become one of the main products in China’s wild edible mushroom commercial market [8].

The bacterial community in the soil has a major influence on the development of ectomycorrhizal symbionts. Bacteria can be selectively transferred or aggregated between soil, mycorrhiza, host, and substrates and contribute to yield, quality, disease resistance, mycelial growth, and mycorrhizal formation [9,10,11,12]. These organisms are also an important factor affecting their spread and development in the wild and maintaining the stability of mycorrhizal microecosystems. For example, α-amorphophilus bacteria and pink yeasts are associated with the ascospores of *T. aestivum* and *T. borchii* Vittad. and may be involved in synthesizing the unique aromatic odor of ascospores [13,14]. Additionally, the slow-growing *Rhizobium* spp., colonized in Italy, and white truffles can increase the absorption of nutrients through nitrogen fixation [15,16].

Soil bacteria in matsutake fruiting body-producing sites can influence the exchange of materials between the ectomycorrhizas and the host plant and also play a key role in the growth and development of mycelium, as well as the formation of matsutake fruiting bodies [9,10,17]. Navarro-Ródenas, A et al. [18,19] obtained a large number of multifunctional, highly active bacterial and actinomycete strains from truffle pond soil through isolation, culture, and functional identification, whose role is to promote the mycorrhization of truffles, stimulate growth, and inhibit pathogenic bacteria. Guo Xia et al. [20,21,22] studied *Bacillus cereus*, *Paenibacillus alvei*, *Pseudomonas extremaustralis*, and other mycorrhizal helper bacteria (MHB) from the bottom of *Boletus edulis*, and *Suillus luteus* bacterial ponds, which can significantly improve mycelial growth, mycorrhizal synthesis, and growth of host seedlings. These MHB can be further used as microbial fertilizers for porcini mushroom ponds. The above research provided theoretical foundations, technical guidance, and new ideas for the artificial cultivation and fine management of valuable mycorrhizal edible fungi, such as *Tricholoma matsutake*, truffles, and *Boletus*.

Although significant progress has been made in cultivating *L*. *hatsudake* mycorrhizal seedlings and establishing plantations [1,2,3], the problems faced by *L*. *hatsudake* plantation producers remain unanswered. For example, the distribution of mycelium and mycorrhiza of *L. hatsudake* in plantation soils, their relationship with bacterial communities, and whether there are bacterial species that can contribute to the formation and development of mycorrhizal fungi in *L*. *hatsudake* remain unknown.

A 6-year-old plantation of *L*. *hatsudake*-Masson Pine mycorrhizal seedlings was the object of this study. The soils of the mushroom-producing area of *L. hatsudake*, including the soil of the fungal base, the soil of the mycorrhizal root, and the soil of the control of *L*. *hatsudake*, were collected during harvest in autumn. High-throughput sequencing technology was used to analyze the composition and diversity of the bacterial community and the relationship between *L*. *hatsudake* and the main bacterial community in the soil. This study is expected to lay the foundations for developing *L*. *hatsudake* mycorrhiza, fruiting body formation, and the interaction mechanism of rhizosphere–soil microorganisms in the wild.

## 2. Research Materials and Methods

### 2.1. Overview of Plantation

The sample collection site was located within an *L. hatsudake* plantation established by our team, located in Wufeng Village, Puji Town, Liuyang City, Hunan Province, China (113°21′–113°31′ E and 27°51′–28°02′ N). The soil type of the collection site was categorized as Felsic metamorphic (MF) group, type as Slate, phyllite (pelitic rocks) according to WRB 2022. The soil pH was 4.77, organic carbon content was 25.24 g/kg, total nitrogen content was 1.26 g/kg, and fast-acting phosphorus content was 7.16 mg/mL. The plantation with the host *Pinus massoniana* was established in the spring of 2014 with a row spacing of 3 m × 3 m. The first *L. hatsudake* production occurred in the autumn of 2016. The climate was characterized by a subtropical monsoon with an even distribution of water and rain, a frost-free period of 271 days, 1656 h of sunshine, and a rainfall of 1552 mm per year. The total area was 10,005 m^2^, the highest point was 126 m above sea level, and the slope was at the west.

### 2.2. Soil Sample Collection

The sampling period for this study was 25 October 2019, when the fruit bodies of *L. hatsudake* produced the most in a year. The plantation was divided into 25 duplicate samples of 20 m × 20 m. Eight non-adjacent samples were selected following a “Z”-shaped pattern. Five mushroom-producing areas and five control areas were randomly selected for the sample collection within each selected sample. The mushroom-producing area refers to the area between the canopy radiation scope and the base of the host tree, where 15 or more fruiting bodies are produced. The control area was the soil area without host tree canopy radiation, and fruiting bodies were never produced. The distance between the adjacent sampling points was approximately 2 m.

The samples were collected from two different habitats: the soil of the fungal base (JT) and the soil of the mycorrhizal root (JG). The collection method of the JT samples at each sampling site consisted of fresh and healthy *L. hatsudake* with hat diameters of 3–6 cm and without insect bites. The fruiting body was placed in a 250 mL sterile container with soil less than 5 cm from the base. The tissue culture bottle was filled with a sterile forceps scoop, numbered, and transported back to the laboratory in a cool box. The soil at the base of the fruiting body was carefully removed with a sterilized surgical blade and mixed in a clean workbench. The JT soil samples were stored at −80 °C as a reserve.

The JG sample collection method was as follows. At first, the host root and mycorrhizal symbiosis were found along the base of the fruiting body. Next, the root segment was cut with a sterile scalpel, placed in a sterilized 250 mL tissue culture bottle, numbered, and transported in a cool box back to the lab. The soil in the root system was obtained with an aseptic brush and mixed in a clean workbench. The JG soil samples were stored at −80 °C as a reserve. The soil of the control area (CK) collection method was as follows. The CK soil samples (0–5 cm) without radiation from the host canopy were collected and mixed in a clean workbench. The CK soil samples were stored at −80 °C as a reserve. Eight samples were obtained from each plot, and 24 soil samples were obtained.

### 2.3. DNA Extraction, Amplification, and Sequencing

#### 2.3.1. Soil Microbial DNA Extraction

The total soil microbial DNA was extracted using a soil DNA extraction kit (MOBIO Power Silo DNA Isolation Kit, BIOGENRO BIOTECHNOLOGY Co., Ltd., Shenzhen, China), and the specific work step followed the standard procedure of the kit instructions. Quantitative DNA nucleic acid analyzers were used to determine the content and purity of DNA, and agarose gel electrophoresis was used to assess the integrity of the DNA samples. The samples with OD260/280 between 1.8 and 2.0 were stored at −80 °C as a reserve. Total soil DNA was extracted using a kit (Hi Pure Soil DNA Kits, Magen Biotech, Guangzhou, China) and purified using a purification kit (UNIQ-10 DNA, Shanghai Biomedicals, Shanghai, China). The quality of DNA was detected by agarose gel electrophoresis (0.5 × Tris Acetate-EDTA buffer) at 10 g/L. The concentration of the extracted DNA was detected using an ultramicro spectrophotometer (Thermo Fisher Scientific, 168 Third Avenue, Waltham, MA, USA). The extracted DNA was diluted to 10 ng/μL and stored at −20 °C as a reserve.

#### 2.3.2. PCR Amplification and Sequencing

Diluted genomic DNA was used as a template. Primers specific to the V4 region of the 16S rRNA gene were used with primer pairs 515F (5′-GTTTCGGTGCCAGCMGCCGCGGGTAA-3′) and 806R (3′-GCCAATGGACTACHVGGGTWTCTAAT-5′). The PCR amplification reaction system comprised 50 μL, including 5 μL PCR buffer (10×), 1 μL MgCl_2_ (1.5 mmol/L), 2 μL dNTP (2.5 mm/L, Taka Ra), 0.25 μL DNA Taq polymerase (2.5 U), and 1 μL primer (1.0 mol/L). A total of 50 ng of soil DNA was used as a template, and ddH_2_O was used to make up to 50 μL. The PCR reaction program included predenaturation at 94 °C for 3 min, denaturation at 94 °C for 40 s, annealing at 56 °C for 60 s, extension at 72 °C for 60 s, and extension at 72 °C for 10 min for 30 cycles. Three technical replicates were performed for each sample. The PCR products of the same sample were mixed and detected by electrophoresis and purified using gel cutting. The samples were sent to Beijing Bemac Biotechnology Co., Ltd. for high-throughput sequencing using the Illumina Hiseq 2500 platform after concentration determination and other procedures.

### 2.4. Sequencing Data Processing

Low-quality reads [23] were cut using Cutadapt (v1.9.1). Then, the samples were separated from the reads based on the barcode. The above processing needs to be processed to remove the chimera sequence. The chimera sequence was recognized using the online software at https://github.com/torognes/vsearch/ (accessed on 24 March 2024) [24] and the species annotation database. Finally, the chimera sequence [25] was removed to obtain the final valid data. The obtained sequencing data were spliced and filtered using COPE and then submitted to the RDP database using QIIME (v1.8.0) with a similarity of 0.97. The operational taxonomic unit (OTU) in each sample was used as the basis for classification and calculation. This study defines an OTU as a sequence group of at least 120 active bases with a base difference of less than 3% (i.e., the resulting analysis). First, the data quality evaluation chart of the two-terminal sequence was displayed to determine the next denoising analysis parameters. The two-terminal sequence was then subjected to quality control, and the feature table was generated using the QIIME dada2 denoise-paired command. The QIIME feature-table filter-features command filters out the features whose total frequency was lower than 5, using the QIIME feature-classifier classesify-sklearn command to compare the filtered sequence with the bacteria database (Silva_132_Release)silva_132_97_16S. The phylogenetic tree was constructed using the QIIME phylogeny align-to-tree-mafft-fasttree command. Then, diversity analysis based on the root tree was performed using the QIIME diversity core-metrics-phylogenetic command to create a layer for bacterial sequences to draw at 20,500.

### 2.5. Community Diversity Analysis

Usearch (v 8.0) was used to cluster Reads at a 97.0% similarity level; the high-quality sequences were clustered using QIIME to obtain OTU and understand the microbial community structure in the samples. A threshold of 0.005% of the sequence number was used to filter OTUs. The simplified OTU list was used for subsequent analyses. The RDP classifier (confidence threshold 0.8) was used to annotate the OTU taxonomy based on the Unite Taxonomy Database. Mothur (v1.30) was used to measure the multidimensional calibration analysis (NMDS). The sample diversity indices, including the richness, Chao1, ACE, Shannon, and Simpson, were calculated using Mothur based on the OTU results. A cluster heat map of soil microbial species was created using Mpson program package for R (v4.2.1). β-diversity analysis based on Bray–Curtis dissimilarity and principal coordinate analysis (PCoA) was used to analyze and plot the heat map of the distance between samples to identify differences in the structure of the detected bacterial community using QIIME.

### 2.6. Analysis of Community Species Differences

The online analysis software LefSe (http://huttenhower.sph.harvard.edu/galaxy, accessed on 26 March 2024) was used to obtain the species diversity between groups based on the OTU abundance matrix. The difference was significant if the linear discriminant analysis (LDA) value was greater than four, and the Kruskal–Wallis rank sum test value was lower than 0.05.

### 2.7. Molecular Ecological Network Construction and Analysis

The OTU data obtained from high-throughput sequencing were standardized, and the Pearson correlation matrix was constructed after uploading the data to the Molecular Ecological Network Analyzes Pipeline (MENAP) website. The molecular ecological networks of microbial communities from 3 treatments (8 replicates per treatment) were built using standard parameters to show the interactions between soil microbial communities in different plantation habitats based on random matrix theory (RMT). The major steps were the same as Ling et al. as follows: a relative abundance matrix, a matrix of soil variables, and an OTU annotation file were prepared in the formats instructed by the pipeline; the relative abundance matrix was submitted for network construction, a cut-off value (similarity threshold, st) for the similarity matrix was automatically generated using default settings, and a link between a pair of OTUs was assigned when the Pearson’s correlation between their RAs exceeded this threshold value; calculations were performed to determine the “global network properties”, the “individual nodes’ centrality”, and the “module separation and modularity”, where a module (or cluster) was a group of nodes more densely connected to each other than to nodes outside the group and modularity was a value measuring how well a network was divided into modules; the “output for Cytoscape visualization” was run in “greedy modularity optimization mode”, and the files were generated for network graph visualization with the Gephi interactive platform; the command “randomize the network structure and then calculate network” was run, enabling the comparison of properties of random and empirical networks [26,27]. Species with a relative abundance of >0.05% were conserved to construct the JT, JG, and CK soil bacterial networks, which were visualized using Gephi (v 0.9.7) [28] based on the results of genus-level sequencing.

## 3. Results and Analysis

### 3.1. Analysis of Soil Bacterial Community Structure in the L. hatsudake Plantation

The 24 soil samples collected from the *L. hatsudake* plantation were sequenced using the Illumina Miseq platform, and 1796620 clean reads were obtained using quality control and splicing. Usearch (v 8.0) was used to cluster Reads at a 97.0% similarity level, and 2056 OTUs were obtained from 24 samples. Among these, 1985 bacterial OTUs were common in the three sites of the *L. hatsudake* plantation. The JT sample had the largest number of bacterial OTUs (2046) and one unique OTU. This unique cluster included Bacteria_Proteobacteria_Deltaproteobacteria_Myxococcales_BIrii41, and uncultured_bacterium_f_BIrii41. The JG sample had the next highest number of bacteria OTUs (2030), and the CK sample had the lowest number of bacteria OTUs (2018). There were no unique OTUs in the JG or CK samples. These data indicate that the composition of OTUs in different types of plantations was different and that the JT soil had more OTUs and unique OTUs. 

According to the species annotation results, bacteria were detected in this study, including 28 phyla, 74 orders, 161 orders, 264 families, 498 genera, and 546 species. The three types of samples contained similar phyla species based on the horizontal distribution histogram of phyla and the composition table of the main bacterial community phyla (Figure 1, Table 1). Proteobacteria and Acidobacteria were the main groups of soil bacteria in the *L. hatsudake* plantation during the mushroom-producing stage. The relative abundance of Proteobacteria in JT (49.44%) was significantly higher than that in JG (30.98%) and CK (25.19%). The relative abundance of Acidobacteria in the JT, JG, and CK samples was 23.79%, 21.54%, and 25.43%, respectively, with no significant difference between the three types. The relative abundance of Actinobacteria, Chloroflexi, and Gemmatimonadetes was significantly lower in JT than in CK and JG, and the relative abundance of Proteobacteria in the JT samples was significantly higher than that in CK and JG.

There was little difference in the species of soil bacterial communities in the three types of samples (JT, JG, and CK) in the *L. hatsudake* plantation based on the horizontal distribution histogram of genera and the horizontal composition table of the main bacterial communities (Figure 2, Table 2). Still, there was one considerable difference in the content of the soil bacterial communities. There were 10 groups according to the bacterial genus: *Acidothermus*, *Bradyrhizobium*, *Burkholderia_Caballeronia_Paraburkholderia*, *Candidatus_Solibacter*, *Candidatus_Xiphinematobacter*, *Granulicella*, *uncultured_bacterium_c_AD3*, *uncultured_bacterium_o_Acidobacteriales*, *uncultured_bacterium_o_Elsterales*, and *uncultured_bacterium_o_Sub Group_2*. Among them, the relative abundance of *Burkholderia_Caballeronia_Paraburkholderia*, *Acidothermus*, *Bradyrhizobium*, *Candidatus_Xiphinematobacter*, and *Granulicella* was significantly higher in mushroom-producing areas (JT and JG) than in control areas (CK). These species can be related to the mycorrhizal and fruiting body development of *L. hatsudake*.

Additionally, some genera or species were found with a relative abundance lower than 0.1% in mushroom-producing areas (JT and JG) but not in the CK samples, such as *Crossiella*, *Polyangium*, *Sorangium*, *Terriglobus*, *Mitsuaria*, *Cystobacter*, *Dactylosporangium*, *Pedobacter*, *Crossiella*, *Luteibacter_yeojuensis*, and *Streptomyces_sp._S30*. The unique genera or species in JT were *Duganella*, *Geobacter*, *Dechloromonas*, *Paludibaculum*, *Minicystis*, *Monodopsis_sp._MarTras21*, *Solitalea*, *Roseimicrobium*, *Cellulomonas*, *Sharpea*, and *Stenotrophomonas_maltophilia*. The unique genera or species in the JG samples were *GOUTA6*, *Clostridium_sensu_stricto_5*, *Noviherbaspirillum*, *Oryza_sativa_Indica_Group_long-grained_rice*, *Candidatus_Uzinura*, and *Rubellimicrobium*.

### 3.2. Analysis of Soil Bacterial Diversity in the L. hatsudake Plantation 

Table 3 shows no significant differences between CK and JG samples in features, and Simpson or Shannon indices. Still, the CK and JG samples were significantly higher than the corresponding values in the JT samples. The results showed that the α-diversity of the bacterial community in JT was lower than that in CK and JG.

The PCoA results (Figure 3a) showed that the eight replicates of the JT soils were separated on the axes from the soil in CK and JG. Axes 1 and 2 explained 34.41% and 19.76% of all variables, respectively, and the cumulative variance contribution of the two principal components was 54.17%. The cluster heatmap of species at the genus level (Figure 3b) shows that most of the eight repeats of JT can be grouped into clusters. However, this cluster was separated from eight repeated JG and CK sample treatments. These data showed that the microbial community structure of soil JT was different from that of JG and CK. 

### 3.3. Differences in Soil Bacterial Communities at Different Types in the L. hatsudake Plantation 

The LefSe analysis (screening criteria: *p* ≤ 0.05, LDA score ≥ 4) was conducted to obtain the LefSe purification branch map and the LDA value distribution histogram (as shown in Figure 4). These analyses were based on the bacterial community composition of soil samples collected from different locations of the *L. hatsudake* plantation. There was a significant difference in the species of biomarker bacteria in various samples from the *L. hatsudake* plantation. There were 24 bacterial biomarker species in the JT group, including Proteobacteria, Betaproteobacteriales, Acidobacteriales, Burkholderiaceae, *Burkholderia_Caballeronia_Paraburkholderi*, Gammaproteobacteria, *uncultured_bacterium_Burkholderia_Caballeronia_Paraburkholderi*, Acidobacteriaceae_Subgroup_1, Alphaproteobacteria, *Granulicella*, *uncultured_bacterium_g_Granulicella*, *Acidipila*, *uncultured-bacterium-Acidipila*, *Aliidongia*, *Aliidongia_dinghuensis*, Elsteraceae, Acetobacteraceae, Acetobacterales, Caulobacterales, Caulobacteraceae, uncultured-bacterium-Caulobacteraceae, and uncultured_bacterium_f_ Acetobacteraceae. There were 11 bacterial biomarker species in the JG group, including Bacteroidetes, *Ktedonobacteria*, Bacteroidia, Ktedonobacterales, Bacteroidales, Acidothermaceae, *Acidothermus*, and Frankiales. There were 15 bacterial biomarker species in the CK group, including Chloroflexi, Bacilli, Lactobacillales, *Streptococcus*, *Streptococcus*, Subgroup_2, and AD3. All of the above species play a very important role in influencing the differences in the bacterial community composition of the *L. hatsudake* plantation.

### 3.4. Characterization of the Molecular Ecological Network Structure of Soil Bacteria in the L. hatsudake Plantation 

The molecular ecological networks of soil bacterial communities in the three types of samples were constructed using Spearman rank correlation analysis based on high-throughput sequencing data (Figure 5, Table 4). After analyzing the data of soil bacteria in plantations, the JT, JG, and CK samples retained 911, 929, and 1103 OTU nodes, respectively. The main characteristic parameters were calculated to describe the overall structure of the network (Table 4) by constructing the molecular ecological network of the bacterial community. The automatically determined thresholds of the three networks were very similar and were 0.94, 0.95, and 0.94, respectively. The average path length and average aggregation coefficient of the molecular ecological networks were larger than the corresponding values of the random networks, and the R^2^ of the topological distributions of the three networks were 0.946, 0.908, and 0.912, respectively, which corresponded to the power-law.

These results suggest that the network constructed in this study was scale-free, small-world, and modular and could be used to study the interrelationships among bacteria. Further analysis shows (Table 4) that the size of the total number of nodes in the bacterial network was CK (1103) > JG (929) > JT (911). The size of the total number of edges was CK (2067) > JT (1508) > JG (1210). The average number of connections was CK (9.96) > JG (8.56) > JT (5.55). The average path distance was JT (3.65) > JG (2.64) > CK (2.18). The network structure of the mushroom-producing areas (JT and JG) was simpler than that of the control areas (CK). At the same time, the extent of the bacterial network and the complexity of the relationship between the bacteria in mushroom-producing areas were smaller than in control areas (CK). The bacterial community structure in JT was small, and that from JG samples was well stabilized and less susceptible to external environmental disturbances than CK.

The results of the phylum-level co-occurrence network analysis with the three samples were consistent with those of the topology analysis, which showed the network differences between the three samples more intuitively. Furthermore, the species relationship in CK was predominantly positively correlated (59.6%). The species relationship in JT was predominantly positively correlated (52.5%). However, the species relationship in JG was mainly negatively correlated (53.1%), probably due to other mycorrhizal fungi in the mycorrhizal interstices. The mycorrhiza formed a competitive relationship with *L. hatsudake*, and the respective mycorrhizal fungi had different species of beneficial bacteria that entered a competitive relationship with each other.

The network graph constructed based on the OTUs of soil bacteria consisted of 488 nodes and 5154 edges. Proteobacteria, Firmicutes, and Actinomycetes occupied the main nodes. The network graph contained seven modules, of which the main modules accounted for 44.47% of the total number of nodes. A total of 98.29% of the connections in the network diagram were positively correlated, indicating an adequate cooperative relationship between the soil bacteria in the *L. hatsudake* plantation.

The correlation between *L. hatsudake* and the bacteria in the 30 highest relative abundances in plantation soils was analyzed based on data from the literature [3], as shown in Figure 5c. *L. hatsudake* was highly significantly and positively correlated with *Burkholderia_Caballeronia_Paraburkholderia*, *Granulicella*, and *Acidipila*, and significantly and positively correlated with *Mucilaginibacter*. *L. hatsudake* was highly and significantly negatively correlated with uncultured_bacterium_c_AD3, *Bifidobacterium*, and *uncultured_bacterium_f_Gemmataceae*, and significantly negatively correlated with uncultured_bacterium_f_JG30-KF-AS9 and uncultured_bacterium_f_Gemmataceae.

## 4. Discussion

This study used high-throughput amplicon sequencing technology to obtain data on bacterial species in soil. Data analysis software R (v 4.2.1)was used to determine the community structure, diversity, and ecological network properties of bacteria in the mushroom-producing areas, including JT and JG, and the control area (CK) of the *L. hatsudake* plantation. The structure and diversity of the soil bacterial community differed at different locations in mushroom-producing and control areas.

This study detected 28 phyla, 74 classes, 161 orders, 264 families, 498 genera, and 546 species of soil bacteria in the *L. hatsudake* plantation. Proteobacteria and Acidobacteria were the main groups of soil bacteria in the *L. hatsudake* plantation during harvest. This is similar to the growth microenvironment of other mycorrhizal edible mushrooms, such as *Boletus edulis*, *Boletus magnificus*, *Tylopilus*, and *Truffle*.

There were more Proteobacteria and Acidobacteria in the soil of these mushroom-producing sites [29,30,31,32], and the dominant genera were *Rhizobium*, *Pseudoxanthomona*, *Burkholderia*, and *Pseudomonas*, which are common MHB [14,33]. In this study, the relative abundance of Proteobacteria was found to be significantly higher in JT than in CK and JG, including *Burkholderia_Caballeronia_Paraburkholderia*, *Acidothermus*, *Bradyrhizobium*, *Candidatus_Xiphinematobacter*, and *Granulicella*. Other dominant genera showed significantly higher relative abundance in mushroom-producing areas (JT and JG) than in control areas (CK), where MHB may have existed.

In this study, we found that the α-diversity of soil bacteria in the JT of the *L. hatsudake* plantation was significantly lower than that in the JG and CK and that the β-diversity was significantly different from that in the JG and CK. This result was consistent with the distribution of the diversity of the soil fungal community in the plantation being similar to the diversity distribution of the soil fungal community in the plantation [3]. The reason may be that *L. hatsudake* dominates in JT soils. This can enrich the bacterial community, which is beneficial for self-growth but repels bacterial species that are unfavorable for self-development, creating the ecological superiority of specific flora.

This result is similar when studying other species of mycorrhizal edible mushrooms, such as Truffles, *Porcini*, *Boletus*, *Floccularia luteovirens*, and MHB, which can be isolated from the soil at mushroom-producing sites. Xing et al. [34] revealed that the diversity of soil bacterial communities on the fungal ring of yellow–green annelid fungi was lower than that of soil bacterial communities in the fungi-less soil outside the ring in Qinghai Province. *Achromobacter*, *Bacillus bacillus*, *Pseudomonas pseudomonas*, and *Stenotrophomonas* were successfully isolated from a group of yellow–green curly mushrooms [35], which were able to promote not only the mycelial growth of *Floccularia luteovirens*, but also the formation of mycorrhizal fungi with *Trifolium repens*.

*Burkholderia_Caballeronia_Paraburkholderia*, a bacterial marker species in the JT samples of mushroom-producing areas, was positively correlated with *L. hatsudake*. This bacterial marker has growth-promoting properties, such as nitrogen fixation, phosphorus release, and plant hormone production [36], promoting mycelial growth and mycorrhizal synthesis [9,10]. Furthermore, *Granulicella* and *Acidipila*, the landmark bacterial species, were found in JT samples and showed a highly significant positive correlation with *L. hatsudake*. Unique genera or species *Duganella*, *Geobacter*, *Dechloromonas*, *Paludibaculum*, *Minicystis*, *Monodopsis_sp._MarTras21*, *Solitalea*, *Roseimicrobium*, *Cellulomonas*, *Sharpea*, and *Stenotrophomonas_maltophilia* have been observed for the first time. However, whether these species belong to the exclusive MHB of *L. hatsudake* still requires further research.

The ecological relationship network of the bacterial community in the JT and JG samples from the *L. hatsudake* plantation became smaller, less complex, more stable, and less susceptible to disturbances from the external environment than in the CK samples. This result is similar to research results on the network relationship of the fungal community in plantations; apart from this, the structure of the bacterial community in the plantation area of *L. hatsudake*. is more complex than that of fungi. Similarly, *L. hatsudake*. can selectively enrich some soil fungi and bacteria, similar to ecological filtration, for its own growth and development [3].

Dong Liu et al. [37] found that the diversity, uniformity, richness, and complexity of the network of bacterial communities in the soil of Indian truffle plantations gradually decreased from empty soil to substrate-adherent soil-substrate epidermis to the substrate interior. Similar to other studies, the bacterial networks studied by these researchers were predominantly and positively connected. *Lactarius hatsudake* also selectively enriches some soil bacteria, such as truffles, similar to ecological filtering for its growth and development.

In this study, only three different areas of soil bacterial diversity were discovered in the *L. hatsudake*-Masson Pine plantation. Further studies on the diversity of fungi and protozoa in fungal ponds and the effects of environmental factors on the microbial diversity of ponds are needed to provide complete information about ways to maintain microbial diversity.

## 5. Conclusions

The bacterial community structure in three different regions of the *L. hatsudake* plantation was analyzed using high-throughput sequencing. The results showed numerous bacterial species in the plantation area, including 28 phyla, 74 classes, 161 orders, 264 families, 498 genera, and 546 species. Additionally, mycorrhizal helper bacteria (MHB) of *L. hatsudake* can occur predominantly in mushroom-producing areas. This study speculated that *L. hatsudake* could enrich the bacteria beneficial for its growth and create the ecological advantage of certain flora. Therefore, the bacterial diversity in the JT samples was significantly lower than in the JG and CK samples. The results showed that the extent and complexity of the bacterial ecological relationship network were lower in the growing areas than in the non-growing areas. The results of this study provide some insights into the mycorrhizal growth-promoting bacteria of *L. hatsudake* and the microbial function of mycorrhizal soil.

## Figures and Tables

**Figure 1 microorganisms-12-01376-f001:**
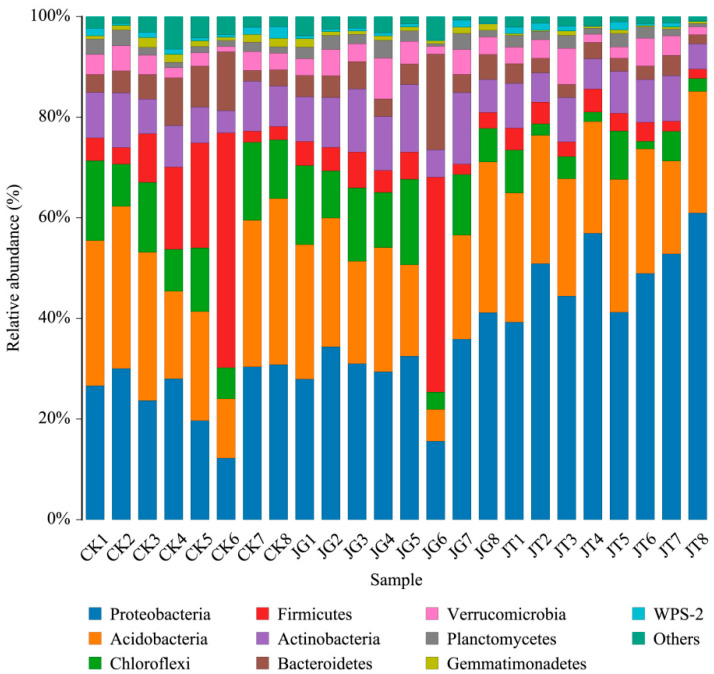
Relative abundance of soil bacterial phylum levels in three types of *L. hatsudake* plantations.

**Figure 2 microorganisms-12-01376-f002:**
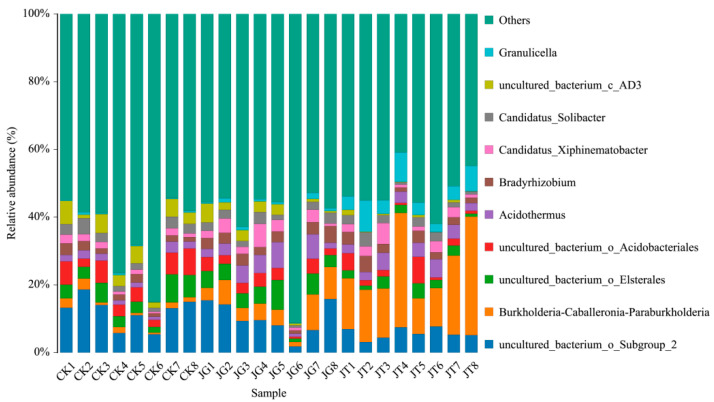
Relative abundance of soil bacterial genera in three types of *L. hatsudake* plantations.

**Figure 3 microorganisms-12-01376-f003:**
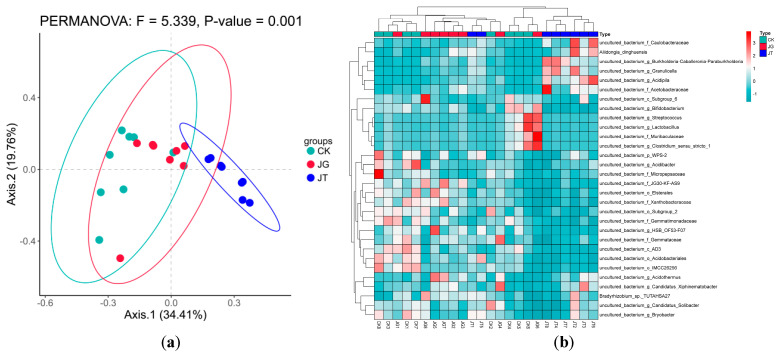
(**a**) Heatmap of PCoA analysis of soil bacteria in different areas of *L. hatsudake* plantation and (**b**) clustering analysis of species abundance.

**Figure 4 microorganisms-12-01376-f004:**
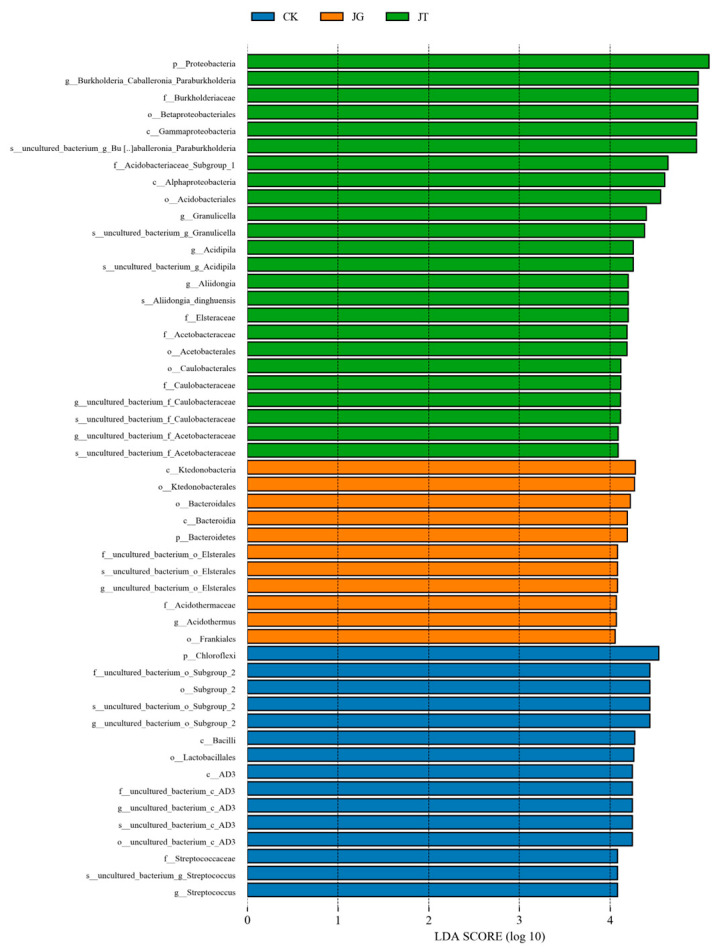
LEfSe histogram of the differentially functional modules at a logarithmic LDA score >4.0 (absolute) of soil bacterial community in *L. hatsudake* plantation at harvesting stage.

**Figure 5 microorganisms-12-01376-f005:**
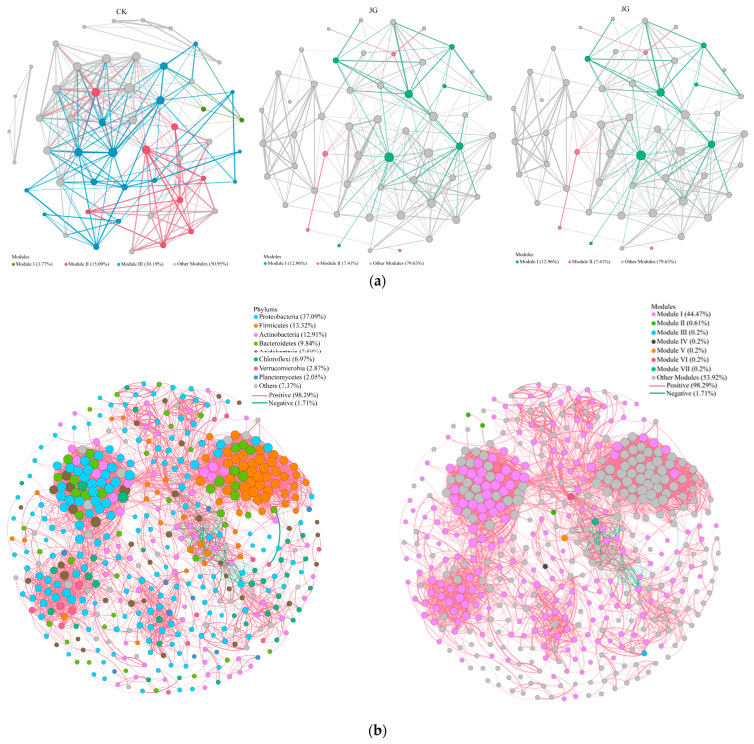
Co-occurrence networks of soil bacterial communities in three different areas of the *L. hatsudake Tanaka* plantation area: (**a**) modular diagrams of the three different treatment groups; (**b**) phylum-level network diagrams and phylum-level modular diagrams; and (**c**) network diagrams of *L. hatsudake Tanaka* with the top 30 bacterial species in terms of relative abundance.

**Table 1 microorganisms-12-01376-t001:** Relative abundance of soil bacterial phylum levels in three types of *L. hatsudake* plantations.

Phylum	CK (%)	JG (%)	JT (%)
Acidobacteria	25.43 ± 7.67 a	21.54 ± 7.26 a	23.79 ± 2.55 a
Actinobacteria	8.06 ± 2.00 ab	10.21 ± 3.17 a	7.55 ± 1.64 b
Bacteroidetes	5.96 ± 3.43 a	6.16 ± 5.24 a	2.98 ± 0.74 a
Chloroflexi	11.58 ± 3.58 a	11.24 ± 4.63 a	4.61 ± 3.13 b
Firmicutes	13.28 ± 15.15 a	9.28 ± 13.59 a	3.42 ± 1.06 a
Gemmatimonadetes	1.26 ± 0.50 a	0.99 ± 0.37 a	0.46 ± 0.24 b
Planctomycetes	1.79 ± 0.84 a	2.21 ± 0.99 a	1.84 ± 0.73 a
Proteobacteria	25.19 ± 6.46 b	30.98 ± 7.12 b	49.44 ± 7.55 a
Verrucomicrobia	3.19 ± 1.25 a	4.33 ± 1.91 a	3.62 ± 1.95 a
WPS-2	1.11 ± 0.63 a	0.59 ± 0.38 a	0.85 ± 0.57 a

Note: CK: soil of the control area, JG: the soil of the mycorrhizal root, JT: soil of the base of the mushroom. Different lowercase letters indicate significant differences between treatments (*p* < 0.05). Values are means ± standard deviation, the same as below.

**Table 2 microorganisms-12-01376-t002:** Relative abundance of soil bacterial genera in three types of *L. hatsudake* plantations.

Genus	CK (%)	JG (%)	JT (%)
*Acidothermus*	1.91 ± 0.74 b	4.23 ± 2.51 a	3.66 ± 1.21 a
*Bradyrhizobium*	2.01 ± 0.81 b	3.10 ± 1.13 a	2.72 ± 1.20 a
*Burkholderia_Caballeronia_Paraburkholderia*	1.58 ± 1.01 c	5.67 ± 3.12 b	19.85 ± 9.76 a
*Candidatus_Solibacter*	2.67 ± 1.14 a	2.24 ± 0.92 a	2.18 ± 1.14 a
*Candidatus_Xiphinematobacter*	1.54 ± 0.71 b	2.99 ± 2.01 a	2.58 ± 1.72 a
*Granulicella*	0.35 ± 0.23 b	0.83 ± 0.49 b	5.44 ± 2.62 a
*uncultured_bacterium_c_AD3*	4.02 ± 2.03 a	2.41 ± 1.75 b	0.48 ± 0.52 c
*uncultured_bacterium_o_Acidobacteriales*	4.98 ± 2.21 a	3.02 ± 1.30 ab	2.58 ± 2.56 b
*uncultured_bacterium_o_Elsterales*	4.55 ± 2.16 a	4.83 ± 2.20 a	2.54 ± 1.17 b
*uncultured_bacterium_o_Subgroup_2*	12.03 ± 4.52 a	10.11 ± 4.84 a	5.69 ± 1.58 b

Different lowercase letters within the same row indicate a significant difference between treatments (*p* < 0.05).

**Table 3 microorganisms-12-01376-t003:** Alpha diversity index of soil bacterial community at different types of *L. hatsudake* plantation.

Sample ID	Clean Reads	Feature	Simpson	Shannon
CK	78,114 ± 101.88 a	1383 ± 37.57 a	0.99 ± 0.001 a	8.29 ± 0.13 a
JG	77,878 ± 65.08 b	1371 ± 26.28 a	0.99 ± 0.002 a	8.12 ± 0.10 a
JT	77,803 ± 60.43 b	13,047 ± 35.19 b	0.97 ± 0.006 b	7.19 ± 0.25 b

Different lowercase letters within the same column indicate a significant difference between treatments (*p* < 0.05).

**Table 4 microorganisms-12-01376-t004:** Topology of molecular ecological networks and stochastic molecular ecological networks of soil bacterial communities in *L. hatsudake* plantations.

	Index	Bacterial		
		CK	JG	JT
Empirical	RMT Threshold	0.940	0.950	0.940
	Total nodes	1103	929	911
	Totaledges	2067	1210	1508
	R^2^ of power-law	0.912	0.908	0.946
	Average degree (avgK)	9.96	8.56	5.55
	Average clustering coefficient (avgCC)	0.178	0.128	0.136
	Average path distance (GD)	2.18	2.64	3.65
	Positive edges	1232 (59.6%)	567 (46.9%)	792 (52.5%)
	Negative edges	835 (40.4%)	643 (53.1%)	716 (47.5%)
Randomized	Average clustering coefficient (avgCC)	0.031 ± 0.003	0.003 ± 0.002	0.004 ± 0.001
	Average path distance (GD)	5.391 ± 0.031	6.911 ± 0.113	5.711 ± 0.049

## Data Availability

The data of this study are available from the corresponding author upon reasonable request.

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
