# Peer review of "The Soil Bacterial Community Structure in a Lactarius hatsudake Tanaka Plantation during Harvest"

_microorganisms, 2024, doi:10.3390/microorganisms12071376_

Round 1

Reviewer 1 Report

Comments and Suggestions for Authors

The Soil Bacterial Community Structure in a Lactarius hatsudake Tanaka Plantation During Harvest

General remarks:

A really interesting work. I have to praise the authors for the idea, but also for the text, which is really professionally written. I would have a few objections, which I will list below:

Page 3, Line 113 - JG or JT? In this paragraph you wrote about JT so it's a bit confusing.

Page 3, Line 119 - Give an explanation for the abbreviation CK.

Page 3, Lines 125, 126 - Specify the DNA kit manufacturer in detail.

Page 3, Lines 130, 131 - Specify the kit manufacturer in detail.

Pages 8 and 9, Figure 4 - Very poor resolution of Figures 4a and 4b. Correct it.

Page 9, Line 301 - Table 4?

Page 9, Line 304 - Table 4?

Page 10, Line 313 - Table 4?

Page 10, Line 315 - 9.86 is the result from Table 4.

References - List the references according to the instructions of the journal. Also, 12 out of 35 literature references are in Chinese, so it is impossible to check them and some could not even be found. If they are able, it would be good if the authors replace them with some more accessible references.

General opinion:

I suggest that the paper be considered for publication after the proposed revision.

Best Regards,

Reviewer

Author Response

Response to Reviewer 1 Comments

Point 1: Page 3, Line 113 - JG or JT? In this paragraph you wrote about JT so it's a bit confusing.

Response 1: Sorry for the confusing description, we have corrected the wrong place in the revised manuscript as follows: “The JT soil samples were stored at ï¹£80°C as a reserve” (Line 116).

Point 2: Page 3, Line 119 - Give an explanation for the abbreviation CK.

Response 2: Thank you very much for your question. We have explained the abbreviation CK in the revised manuscript as follows: “The soil of the non-mushroom-producing area (CK) collection method was as follows. The CK soil samples (0-5 cm) without radiation from the host canopy were collected and mixed in a clean workbench” (Line 122-124).

Point 3: Page 3, Lines 125, 126 - Specify the DNA kit manufacturer in detail.

Response 3: I am very sorry for the confusion. We have specified the DNA kit manufacturer in the revised manuscript as follows: “The total soil microbial DNA was extracted using a soil DNA extraction kit (MO-BIO Power Silo DNA Isolation Kit, BIOGENRO BIOTECHNOLOGY CO., LTD)” (Line 130).

Point 4: Page 3, Lines 130, 131 - Specify the kit manufacturer in detail.

Response 4: Thank you very much for your suggestion, we have specified the kit manufacturer in the revised manuscript as follows: “Total soil DNA was extracted using a kit (Hi Pure Soil DNA Kits, Magen Biotech) and purified using a purification kit (UNIQ-10 DNA, Shanghai Biomedicals, China)” (Line 135).

Point 5: Pages 8 and 9, Figure 4 - Very poor resolution of Figures 4a and 4b. Correct it.

Response 5: Thank you very much for your suggestion. We have replaced Figures 4a and 4b with higher-resolution images.

Point 6: Page 9, Line 301 - Table 4?

Response 6: We apologize very much for our carelessness, and have corrected the wrong place in the revised manuscript as follows: “The molecular ecological networks of soil bacterial communities in the three types of samples were constructed using Spearman rank correlation analysis based on high-throughput sequencing data (Fig. 4, Table 4)” (Line 306).

Point 7: Page 9, Line 304 - Table 4?

Response 7: We apologize very much for our carelessness, and have corrected the wrong place in the revised manuscript as follows: “The main characteristic parameters were calculated to describe the overall structure of the network (Table 4) by constructing the molecular ecological network of the bacterial community” (Line 309).

Point 8: Page 10, Line 313 - Table 4?

Response 8: We apologize very much for our carelessness, we have corrected the wrong place in the revised manuscript as follows: “Further analysis shows (Table 4) that the size of the total number of nodes in the bacterial network was CK (1103) > JG (929) > JT (911)” (Line 318).

Point 9: Page 10, Line 315 - 9.86 is the result from Table 4.

Response 9: We apologize very much for our carelessness. After checking, it was found that there was an error in the data in Table 4, and we have corrected it. (Line 348)

Point 10: References - List the references according to the instructions of the journal. Also, 12 out of 35 literature references are in Chinese, so it is impossible to check them and some could not even be found. If they are able, it would be good if the authors replace them with some more accessible references.

Response 10: Thank you very much for your suggestion. We have replaced all but 20-22 of the Chinese literature references with more accessible English references. (Line 463)

Reviewer 2 Report

Comments and Suggestions for Authors

The manuscript entitled, 'Soil Bacterial Community Structure in a Lactarius hatsudake Tanaka Plantation During Harvest by Shen et al., (2024) is an interesting study. I have only one concern regarding the paper that authors should clarify. The research objective of the manuscript remains unclear. The paper talks about the bacterial community structure in response to mushroom plantation. However, in the abstract, the authors mentioned that plantation of mushroom was the research objective. This is misleading and authors should clarify this.

Author Response

Response to Reviewer 2 Comments

Point 1: The manuscript entitled, 'Soil Bacterial Community Structure in a Lactarius hatsudake Tanaka Plantation During Harvest by Shen et al., (2024) is an interesting study. I have only one concern regarding the paper that authors should clarify. The research objective of the manuscript remains unclear. The paper talks about the bacterial community structure in response to mushroom plantation. However, in the abstract, the authors mentioned that plantation of mushroom was the research objective. This is misleading and authors should clarify this.

Response 1: I am very sorry for the confusion. We have changed the purpose of the manuscript's abstract to “In this study, the plantation of L. hatsudake during the harvest period was taken as the research object, and this article explores which bacteria in the soil contribute to the production and growth of L. hatsudake” (Line 12-14).

Reviewer 3 Report

Comments and Suggestions for Authors

Review

The authors in the manuscript submitted for review entitled "The Soil Bacterial Community Structure in a Lactarius hatsudake Tanaka Plantation During Harvest" focused on determining the species composition of the bacterial community in the soil collected from various locations of the Lactarius hatsudake Tanaka plantation. The work is similar to the one published under the title "Diversity and Network Relationship Construction of Soil Fungal Communities in Lactarius hatsudake Tanaka Orchard during Harvest". After analyzing the text, I have several comments and suggestions:

1. The authors should formulate the aim of the undertaken research more precisely and clearly.

2. Explain what the authors were guided by when choosing the locations of soil sampling for the analysis of the bacterial community composition. In my opinion, the CK designation would be more understandable to the reader as a control, i.e. outside the plantation.

3. Please include more detailed information on the physical and chemical properties of soils such as: soil texture, organic carbon content and total nitrogen, and also present the taxonomic systematics of the soil according to WRB 2022.

4. Please refer to the previously published article in more detail in the discussion  entitled "Diversity and Network Relationship Construction of Soil Fungal Communities in Lactarius hatsudake Tanaka Orchard during Harvest". Engage in a broader discussion in connection with this article.

5. In addition, editorial suggestions for drawings - poorly legible names of bacteria

The article requires correction and additions

Author Response

Response to Reviewer 3 Comments

Point 1: The authors should formulate the aim of the undertaken research more precisely and clearly.

Response 1: Thank you very much for your suggestion. We have made the purpose of the study clear in the abstract. The changes in the manuscript are as follows: “In this study, the plantation of L. hatsudake during the harvest period was taken as the research object, and this article explores which bacteria in the soil contribute to the production and growth of L. hatsudake” (Line 12-14).

Point 2: Explain what the authors were guided by when choosing the locations of soil sampling for the analysis of the bacterial community composition. In my opinion, the CK designation would be more understandable to the reader as a control, i.e. outside the plantation.

Response 2: We are very sorry for the confusion. To study the soil bacterial community composition in different areas within the L. hatsudake plantation, we divided the experiment into three groups, CK(non-mushroom-producing area), JT(soil of the fungal base), JG(the soil of the mycorrhizal root). The CK group was not outside the plantation but in the non-mushroom-producing area outside the base and roots of the distant ionic entities.

Point 3: Please include more detailed information on the physical and chemical properties of soils such as: soil texture, organic carbon content and total nitrogen, and also present the taxonomic systematics of the soil according to WRB 2022.

Response 3: We are very sorry for the confusion, we have added the data related to the physicochemical properties of the soil and the soil type of the collection site to the manuscript as follows: “The soil type of the collection site was slate yellow-red soil and brown-yellow soil. Soil pH was 4.77, organic carbon content was 25.24 g/kg, total nitrogen content was 1.26 g/kg and quick-acting phosphorus content was 7.16 mg/ml” (Line 88-91).

Point 4: Please refer to the previously published article in more detail in the discussion entitled "Diversity and Network Relationship Construction of Soil Fungal Communities in Lactarius hatsudake Tanaka Orchard during Harvest". Engage in a broader discussion in connection with this article.

Response 4: Thank you very much for your suggestion, we refer to the previously published article "Diversity and Network Relationship Construction of Soil Fungal Communities in Lactarius hatsudake Tanaka Orchard during Harvest" and discuss it more extensively concerning this article as follows: “The ecological relationship network of the bacterial community in the JT and JG samples from the L. hatsudake plantation became smaller, less complex, more stable, and less susceptible to disturbances from the external environment than in the CK samples. This result is similar to research results on the network relationship of the fungal community in plantations, apart from this, the structure of the bacterial community in the plantation area of L. hatsudake. is more complex than that of fungi. Similarly, L. hatsudake. can selectively enrich some soil fungi and bacteria, similar to ecological filtration, for its own growth and development [3] ” (Line 410-417).

Point 5: In addition, editorial suggestions for drawings - poorly legible names of bacteria.

Response 5: Thank you very much for your suggestion. We have replaced Figures 3b, 4a, 4b, and 5c with higher-resolution images to higher legible bacterial names.

Reviewer 4 Report

Comments and Suggestions for Authors

The manuscript is reporting valuable data about the microbial, bacterial community structure in two different sites, either is cultivated by mushrooms or not. The up-to-date methodology the high throughout put amplicon sequencing technology were used. Mathematical evaluation proved to be very useful either for analysing the results and to demonstrate its relevance. After those whole genome analysis, the demonstrative figures are showing the real differences among the sites. It is interesting to learn, that the mushrooms are dominating in the structure-management at both soils and in the biodiversity issues. There were detected 28 phyla, 74 classes, 161 orders, 264 families, 498 genera, and 546 species of soil bacteria. Value of the study that we can have information about the so-called non-cultivating microbes (for instance the Verrumicroba) that are still present in the soils. The manuscript is well-written and is demonstrative for the potential readers, furthermore, is convenient for the journal of Microorganisms. It can be suggested for publication on its present form.

Author Response

Thank you very much for your positive comments.

Round 2

Reviewer 3 Report

Comments and Suggestions for Authors

Review

 In the revised version of the manuscript, the authors made corrections mostly in line with my suggestions. However, I have a few more comments:

Response 2: We are very sorry for the confusion. To study the soil bacterial community composition in different areas within the L. hatsudake plantation, we divided the experiment into three groups, CK(non-mushroom-producing area), JT(soil of the fungal base), JG(the soil of the mycorrhizal root). The CK group was not outside the plantation but in the non-mushroom-producing area outside the base and roots of the distant ionic entities.

At this point, I would suggest the designation CK – control

Point 3: Please include more detailed information on the physical and chemical properties of soils such as: soil texture, organic carbon content and total nitrogen, and also present the taxonomic systematics of the soil according to WRB 2022.

Response 3: We are very sorry for the confusion, we have added the data related to the physicochemical properties of the soil and the soil type of the collection site to the manuscript as follows: “The soil type of the collection site was slate yellow-red soil and brown-yellow soil. Soil pH was 4.77, organic carbon content was 25.24 g/kg, total nitrogen content was 1.26 g/kg and quick-acting phosphorus content was 7.16 mg/ml” (Line 88-91).

I asked for the taxonomy of the soil on which the experiment was conducted according to the WRB 2022 taxonomy. Please include the correct taxonomy of this soil in accordance with the WRB guidelines.

IUSS Working Group WRB. 2022. World Reference Base for Soil Resources. International soil classification system for naming soils and creating legends for soil maps. 4th edition. International Union of Soil Sciences (IUSS), Vienna, Austria.

World Reference Base for Soil Resources (obrl-soil.github.io)

Author Response

Point 1: At this point, I would suggest the designation CK – control

Response 1: Thank you very much for your suggestion, we have interpreted CK as a control group as you suggested to make it easier for readers to understand.

Point 2: I asked for the taxonomy of the soil on which the experiment was conducted according to the WRB 2022 taxonomy. Please include the correct taxonomy of this soil in accordance with the WRB guidelines.

Response 2:  Thank you very much for your suggestion, I have provided the soil classification criteria for conducting the experiment under WRB 2022 classification as per your request. 
